# Investigation of Novel Therapeutic Targets for Rheumatoid Arthritis Through Human Plasma Proteome

**DOI:** 10.3390/biomedicines13081841

**Published:** 2025-07-29

**Authors:** Hong Wang, Chengyi Huang, Kangkang Huang, Tingkui Wu, Hao Liu

**Affiliations:** Department of Orthopedics, West China Hospital of Sichuan University, Chengdu 610041, China; 2018141431266@stu.scu.edu.cn (H.W.); 2022324020067@stu.scu.edu.cn (C.H.); huangkangkang@scu.edu.cn (K.H.)

**Keywords:** rheumatoid arthritis, plasma proteome, Mendelian randomization, therapy, drug target

## Abstract

**Background**: Rheumatoid arthritis (RA) is an autoimmune disease that remains incurable. An increasing number of proteomic genome-wide association studies (GWASs) are emerging, offering immense potential for identifying novel therapeutic targets for diseases. This study aims to identify potential therapeutic targets for RA based on human plasma proteome. **Methods**: Protein quantitative trait loci were extracted and integrated from eight large-scale proteomic GWASs. Proteome-wide Mendelian randomization (Pro-MR) was performed to prioritize proteins causally associated with RA. Further validation of the reliability and stratification of prioritized proteins was performed using MR meta-analysis, colocalization, and transcriptome-wide summary-data-based MR. Subsequently, prioritized proteins were characterized through protein–protein interaction and enrichment analyses, pleiotropy assessment, genetically engineered mouse models, cell-type-specific expression analysis, and druggability evaluation. Phenotypic expansion analyses were also conducted to explore the effects of the prioritized proteins on phenotypes such as endocrine disorders, cardiovascular diseases, and other immune-related diseases. **Results**: Pro-MR prioritized 32 unique proteins associated with RA risk. After validation, prioritized proteins were stratified into four reliability tiers. Prioritized proteins showed interactions with established RA drug targets and were enriched in an immune-related functional profile. Four trans-associated proteins exhibited vertical or horizontal pleiotropy with specific genes or proteins. Genetically engineered mouse models for 18 prioritized protein-coding genes displayed abnormal immune phenotypes. Single-cell RNA sequencing data were used to validate the enriched expression of several prioritized proteins in specific synovial cell types. Nine prioritized proteins were identified as targets of existing drugs in clinical trials or were already approved. Further phenome-wide MR and mediation analyses revealed the effects and potential mediating roles of some prioritized proteins on other phenotypes. **Conclusions**: This study identified 32 plasma proteins as potential therapeutic targets for RA, expanding the prospects for drug discovery and deepening insights into RA pathogenesis.

## 1. Introduction

Rheumatoid arthritis (RA) is a chronic autoimmune inflammatory disease primarily affecting synovial joints and multiple extra-articular systems [1]. As of 2020, approximately 17.6 million people worldwide are affected by RA, with its prevalence showing a steady upward trend [2]. RA leads to loss of work capacity and premature death, placing a substantial economic burden on society [3].

RA remains incurable. With advancements in targeted treatments and implementation of the treat-to-target strategy, substantial improvements have been made in RA disease control [4]. However, there are still several limitations to the existing disease-modifying antirheumatic drugs (DMARDs). First, each drug has limited efficacy, with monotherapy achieving disease activity reduction or remission only in up to 40% of RA patients [5]. Consequently, dose adjustments and combination therapies are almost always required, inevitably raising concerns about drug-related toxicities or adverse events. Additionally, relapse is common upon treatment discontinuation [5]. Therefore, identifying novel therapeutic targets to develop more optimal treatments is an urgent and ongoing pursuit.

Plasma proteins, originating from either cellular leakage or active secretion, are closely associated with disease states. Targeting proteins for drug development has proven to be both feasible and promising [6]. Numerous observational studies have explored RA-related biomarkers or potential drug targets based on changes in plasma protein levels [7,8,9]. However, these studies are often limited by research design, methodology, and sample size, resulting in constraints such as a high level of confounding factors and an inability to establish clear causal relationships.

Genetic tools have been widely used to establish links between single nucleotide polymorphisms (SNPs) and human traits. To date, multiple large-scale proteomic genome-wide association studies (GWASs) have identified tens of thousands of sequence determinants affecting the abundance of various plasma proteins, known as protein quantitative trait loci (pQTLs), providing a valuable resource for high-throughput exploration of protein–disease associations. Another tool, Mendelian randomization (MR) analysis, based on GWAS summary statistics, uses genetic variants as instrumental variables (IVs) to simulate a quasi-randomized trial. MR enables effective identification of causal relationships between traits while minimizing reverse causation and confounding effects [10]. As a combination of these approaches, proteome-wide Mendelian randomization (Pro-MR), a causality-testing framework that leverages pQTLs as instrumental variables, has provided important insights into drug target prioritization and the elucidation of disease mechanisms in conditions such as diabetes, colorectal cancer, and multiple sclerosis [11,12,13]. Previous studies have sought to identify novel targets for autoimmune diseases through Pro-MR [14,15], yet they are limited by narrow protein coverage and insufficient validation, characterization, and focused discussion, warranting further refinement.

Herein, we aim to perform a Pro-MR analysis using data from eight proteomic GWASs to comprehensively prioritize plasma proteins that are causally associated with RA risk as potential therapeutic targets. Furthermore, we will apply multidimensional approaches to validate and characterize these prioritized proteins.

## 2. Materials and Methods

The study methodology is outlined in Figure 1. All data sources are either publicly available or accessible upon reasonable request. Details regarding data imputation, quality control principles, ethical approvals, informed consent, and other information are traceable to the original sources. Unless otherwise specified, the analyses were conducted on R platform (ver. 4.2.3, R Foundation for Statistical Computing, Vienna, Austria) with specific packages.

### 2.1. Data Sources of Plasma Proteins and RA

Given the complementary nature and varying statistical strengths of identified associations across different proteomic GWASs, we extracted and integrated data from eight large-scale studies [16,17,18,19,20,21,22,23] (Appendix A) to acquire and refine candidate pQTLs as genetic IVs for plasma proteins. These proteomic GWASs were derived from independent cohorts. Two datasets, one including only cis-pQTLs and the other containing all pQTLs, were constructed, comprising 2634 and 14,247 pQTLs, and associated with 2483 and 5145 unique proteins, respectively. The data integration is detailed in Appendix A.

The discovery and replication datasets for RA were obtained from two GWAS meta-analyses by Ishigaki et al. [24] and Sakaue et al. [25], respectively. Additionally, datasets on trans-ethnic and East Asian RA, seropositive and seronegative RA, juvenile RA, rheumatoid factor and joint damage were included. Specific information about these data sources is provided in Appendix A.

### 2.2. Pro-MR Analysis

Pro-MR analysis was conducted between plasma proteins and RA, based on a batch of univariate two-sample MR analyses. Considering that cis-pQTLs have more direct biological relevance, and the inclusion of trans-pQTLs broadens the scope for identifying potential therapeutic targets, two datasets containing cis-pQTLs and all pQTLs were analyzed separately. Discovery MR leveraged proteins as exposure and the discovery RA dataset as outcome. Various sensitivity analyses were employed to enhance the robustness and reliability of the primary findings. Given multiple testing, Bonferroni correction was applied to adjust significance thresholds to *p* < 2.014 × 10^−5^ (0.050/2483) for the analysis of the cis-pQTLs dataset and *p* < 9.718 × 10^−6^ (0.050/5145) for the analysis of the all-pQTLs dataset. Proteins that survived the adjusted significance thresholds were selected as prioritized proteins for further analysis. Subsequently, replication MR for prioritized proteins was performed using the replication RA dataset. Results from the discovery and replication stages were meta-analyzed to obtain pooled results. Reverse MR was conducted using the discovery RA dataset as exposure and prioritized proteins as outcome to further assess reverse causality. Finally, MR was also used to investigate the consistency of effects of prioritized proteins on different RA-related phenotypes. For these MR and meta-analysis procedures, results were considered statistically significant at *p* < 0.050. All MR analyses in this study were performed using the “TwoSampleMR” package (ver. 0.5.8) [26], and meta-analysis was conducted using the “meta” package (ver. 6.6-0). Further details of the methods are provided in Appendix A.

### 2.3. Colocalization Analysis

Due to potential false-positive findings due to confounding from the linkage disequilibrium (LD) structure of variants, colocalization analysis was performed to evaluate whether the protein and the RA trait shared the same causal variant. For each prioritized protein, comprehensive summary statistics were obtained from the corresponding studies. The RA summary statistics were derived from the discovery RA dataset. Of particular interest is the posterior probability for hypothesis 4 (PPH_4_), with 0.800 > PPH_4_ ≥ 0.500 indicating moderate evidence of colocalization and PPH_4_ ≥ 0.800 indicating strong evidence. The method was detailed in Appendix A. Colocalization analysis was conducted with the “coloc” package (ver. 5.2.3) [27].

### 2.4. Transcriptome-Wide Summary-Data-Based MR (SMR)

SMR analysis was conducted to further validate the causal relationships between prioritized protein-coding genes and RA risk from the transcriptomic perspective. The expression quantitative trait locus (eQTL) most significantly associated with each gene was used as the exposure, with the discovery RA dataset serving as the outcome. Multiple eQTL datasets were included. The significance threshold was set at *p* < 0.050. The Heterogeneity in Dependent Instruments (HEIDI) test was also performed, using multiple SNPs in a region to assess whether gene expression and RA traits shared a causal variant rather than being linked by genetic correlation. A HEIDI test with *p* < 0.010 indicated potential linkage. Detailed methodologies are provided in Appendix A. Both SMR analysis and the HEIDI test were performed using SMR software (ver. 1.3.1) [28].

### 2.5. Reliability Stratification of Prioritized Proteins

Based on the validation evidence from meta-analysis, colocalization, and SMR results, we stratified prioritized proteins into four reliability tiers: Tier 1 (proteins passing all tests), Tier 2 (proteins passing two tests), Tier 3 (proteins passing one test), and Tier 4 (proteins not passing any test).

### 2.6. Protein–Protein Interaction and Enrichment Analyses

Protein–protein interaction (PPI) network was employed to delineate interactions among prioritized proteins and with current drug targets. We identified approved DMARDs from established guidelines and reviews [4,5], annotating their primary target genes based on the literature and DrugBank database (https://www.drugbank.ca [accessed on 8 May 2024]). PPI analysis was conducted with the Search Tool for Recurring Instances of Neighboring Genes (STRING, https://string-db.org [accessed on 23 May 2024]) database, using a minimum interaction score of 0.4.

Additionally, Gene Ontology (GO) and Kyoto Encyclopedia of Genes and Genomes (KEGG) pathway enrichment analysis were performed to characterize the biological functions and pathways associated with prioritized proteins. Enrichment analysis was performed using the “clusterProfiler” package (ver. 4.6.2) [29].

### 2.7. Pleiotropy Assessment

Considering the more complex biological connections of trans-pQTLs compared to cis-pQTLs, we assessed potential pleiotropy mediated by the involved trans-pQTLs to enhance their interpretability, using an approach modified from previous research [30]. The detailed steps were described in Appendix A.

### 2.8. Genetically Engineered Mouse Models

Mouse Genomics Informatics (MGI) database (http://www.informatics.jax.org [accessed on 7 June 2024]) was queried for evidence that the modification of genes encoding prioritized proteins produces phenotypes relevant to RA. MGI compiles data from the primary literature, organized using standardized nomenclature and controlled vocabularies. We recorded information on mouse orthologs for each queried gene and details on all modifying models. Phenotypes associated with each model were parsed, with particular focus on abnormalities related to immune system, skeleton, and mortality/aging.

### 2.9. Cell-Type-Specific Expression Analysis

Protein abundance differs between plasma and tissue. To further investigate cell-type-specific expression patterns of genes encoding prioritized proteins in synovium, we re-analyzed single-cell RNA sequencing (scRNA-seq) data from synovial samples generated by Zhang et al. [31]. The data processing procedures are detailed in Appendix A. All analyses were performed in R using the “Seurat V4” package [32].

### 2.10. Druggability Evaluation

Combining information from DrugBank, Open Targets Platform (https://platform.opentargets.org [accessed on 10 June 2024]), Therapeutic Target Database (TTD, https://db.idrblab.net/ttd [accessed on 11 June 2024]), and a literature review, we evaluated the druggability of prioritized proteins. These databases provide up-to-date and comprehensive information on drugs, drug–gene interactions, gene functions, and more. Relevant drug properties, indications, and development stages were extracted and integrated.

### 2.11. Phenome-Wide MR (Phe-MR) Analysis

Phe-MR analysis aids in revealing the effects of prioritized proteins on a broad range of other phenotypes, uncovering additional therapeutic effects or adverse drug reactions. Phe-MR was based on a batch of univariate two-sample MR analyses. The pQTLs employed as IVs for prioritized proteins were consistent with Pro-MR. Summary statistics for multiple disease phenotypes from a SAIGE GWAS of a UK Biobank cohort including 408,961 individuals were obtained as an outcome (https://www.leelabsg.org [accessed on 28 June 2024]) (Appendix A). After excluding RA and unspecified traits, 1161 binary traits were retained for analysis. The significance threshold was corrected to *p* < 4.307 × 10^−5^ (0.050/1161).

### 2.12. Mediation Analysis

To investigate the regulatory effects of modifiable factors (gut microbiome, body measurements, lifestyle, and dietary patterns) on RA and explore the potential mediating role of prioritized proteins, we conducted a series of MR analyses (Appendix A). Sources of summary statistics for 247 modifiable factors are listed in Appendix A.

## 3. Results

### 3.1. Pro-MR Prioritized 32 Plasma Proteins for RA

The F-statistics for all IVs exceeded 10, confirming their robust statistical strength. In the discovery stage, genetically predicted plasma levels of 23 and 17 proteins showed significant associations with RA risk based on the cis-pQTLs and all-pQTLs datasets, respectively. Integrating these findings yielded 32 unique prioritized proteins, with overlapping proteins across both datasets demonstrating consistent effect directions. Detailed information on these proteins was provided in Table 1. Proteins were reported using the non-italicized forms of corresponding gene symbols as short names throughout the study. Genetic susceptibility to higher plasma levels of ICOSLG, ERBB2, ALDH2, FCGR3A, PAM, TNFAIP3, CD40, MFAP2, BCL2L15, HAPLN4, SUGP1, FCRL3, OLFML3, PADI4, WASL, POLR2F, IGSF11, and ADPGK correlated with increased RA risk, while NFKBIE, CCL21, IL6R, CD28, FCGR2A, FLT3, IFNGR2, SPRED2, CCL19, CELF2, H2AZ1, H2BC21, H2AC25, and H2BC26 showed inverse associations (Figure 2). Notably, WASL, CELF2, H2AZ1, POLR2F, H2BC21, IGSF11, H2AC25, H2BC26, and ADPGK were exclusively trans-associated proteins (Appendix A). Steiger filtering verified the causal directionality from proteins to the RA trait for all associations. No evidence of heterogeneity or pleiotropy was detected in applicable analyses (Appendix A).

Combining results from the discovery and replication stages, 23 proteins (NFKBIE, CCL21, ICOSLG, ALDH2, FCGR3A, IL6R, CD28, FCGR2A, TNFAIP3, CD40, IFNGR2, MFAP2, SPRED2, HAPLN4, FCRL3, CCL19, PADI4, WASL, POLR2F, H2BC21, IGSF11, H2BC26, and ADPGK) passed meta-analysis (Table 1). Reverse MR identified several reverse causality pairs, indicating that genetically predicted increased risk of RA was associated with higher plasma levels of HAPLN4, FCRL3, and IGSF11, and lower levels of OLFML3 (Appendix A).

Prioritized proteins were also associated with other RA-related phenotypes, as shown in Figure 3. All prioritized proteins maintained significant and consistent effects across a trans-ethnic population. With East Asian RA, some proteins were not available, and CCL21, ALDH2, CD28, FCGR2A, PAM, CCL19, CELF2, H2AC25, and H2BC26 no longer exhibited significant causal effects on RA, while BCL2L15 and OLFML3 showed effects opposite to those observed in the discovery stage. For seropositive RA, all prioritized proteins exhibited consistent and significant associations with the discovery stage findings, whereas for seronegative RA, only CD28, IFNGR2, WASL, POLR2F, and ADPGK kept such associations. Further analysis indicated a protective role for CD28 in juvenile RA. Additionally, genetic susceptibility to higher plasma levels of CD28 and WASL was associated with lower and higher levels of rheumatoid factor, respectively. And genetically predicted higher SPRED2 levels were linked to milder joint damage.

### 3.2. Colocalization Analysis Supported Causations Between 14 Prioritized Proteins and RA

Among the 32 causal plasma proteins prioritized by Pro-MR, 14 exhibited genetic colocalization through coloc.abf or coloc.susie, suggesting a high probability of shared causal variants between genetic protein levels and RA risk, independent of linkage effects. Of these, 12 proteins (NFKBIE, ICOSLG, IL6R, CD40, IFNGR2, HAPLN4, SUGP1, FCRL3, PADI4, WASL, POLR2F, and ADPGK) showed strong evidence, while the remaining 2 proteins (ALDH2 and FCGR2A) demonstrated moderate evidence (Appendix A).

### 3.3. SMR Validated 18 Prioritized Proteins of RA from Transcriptive Perspective

Eighteen prioritized protein-coding genes showed significant associations between genetically predicted expression levels in at least one tissue and RA risk, confirmed by the HEIDI test. These included *ICOSLG*, *ERBB2*, *ALDH2*, *FCGR3A*, *FCGR2A*, *PAM*, *TNFAIP3*, *CD40*, *FLT3*, *IFNGR2*, *MFAP2*, *BCL2L15*, *HAPLN4*, *SUGP1*, *FCRL3*, *OLFML3*, *PADI4*, and *IGSF11*. Among them, nine genes (*FCGR3A*, *FCGR2A*, *TNFAIP3*, *CD40*, *FLT3*, *MFAP2*, *HAPLN4*, *FCRL3*, and *OLFML3*) demonstrated expression levels that were associated with RA risk in consistent effect directions across different tissues, aligning with those observed in plasma protein levels. However, four genes (*ICOSLG*, *ERBB2*, *BCL2L15*, and *PADI4*) exhibited completely opposite effects on RA risk to their corresponding plasma proteins across tissues. Notably, some genes, such as *ALDH2*, *PAM*, *IFNGR2*, *SUGP1*, and *IGSF11*, presented tissue-specific heterogeneity in their effects on RA risk. For instance, the effects of *SUGP1*’s expression level on RA aligned with that of its protein in blood samples, but showed opposite effects in other tissues (Figure 4).

### 3.4. Four Reliability Tiers of Prioritized Proteins

Reliability stratification identified eight proteins in Tier 1 (ICOSLG, ALDH2, FCGR2A, CD40, IFNGR2, HAPLN4, FCRL3, and PADI4) and ten in Tier 2 (NFKBIE, FCGR3A, IL6R, TNFAIP3, MFAP2, SUGP1, WASL, POLR2F, IGSF11, and ADPGK), both considered highly reliable. Tier 3 included eleven proteins (CCL21, ERBB2, CD28, PAM, FLT3, BCL2L15, SPRED2, CCL19, OLFML3, H2BC21, and H2BC26), and Tier 4 included three (CELF2, H2AZ1, and H2AC25), with low reliability (Table 1).

### 3.5. PPI Network and Enriched Pathways of Prioritized Proteins

The PPI network revealed interactions between prioritized proteins and existing RA drug targets. Among the prioritized proteins, IL6R is already a target for established DMARDs. NFKBIE, CCL21, ICOSLG, ERBB2, FCGR3A, IL6R, CD28, FCGR2A, TNFAIP3, CD40, FLT3, IFNGR2, MFAP2, FCRL3, CCL19, PADI4, H2BC21, IGSF11, and ADPGK exhibited interactions with at least one existing drug target. In addition, interactions were also identified between prioritized proteins, such as H2AZ1, POLR2F, H2BC21, H2AC25, and H2BC26. However, ALDH2, PAM, BCL2L15, SPRED2, HAPLN4, SUGP1, OLFML3, WASL, and CELF2 were independent of both other prioritized proteins and existing drug targets (Figure 5A).

Enrichment analysis of genes for prioritized proteins revealed their biological roles. GO enrichment identified their involvement in biological processes such as leukocyte, mononuclear cell, and lymphocyte proliferation. These prioritized proteins predominantly localized to the external side of the plasma membrane, protein–DNA complexes, DNA packaging complexes, and nucleosomes. Their molecular functions encompassed protein heterodimerization activity, cytokine receptor binding, immune receptor activity, and structural constituent of chromatin (Figure 5B). KEGG pathways highlighted the functional enrichment of prioritized proteins in systemic lupus erythematosus, neutrophil extracellular trap formation, and cytokine–cytokine receptor interactions (Figure 5C).

### 3.6. Four Trans-Associated Proteins Exhibited Vertical or Horizontal Pleiotropy

Through pleiotropy assessment, nine SNPs that trans-influence plasma levels of 10 prioritized proteins were also associated with the levels 102 unique proteins and 30 unique gene expressions, generating 208 prioritized protein-secondary gene pairs. At least one cis-pQTL or cis-eQTL was significantly associated with 174 secondary genes and was available as an IV. Subsequent MR analyses identified 13 unique secondary genes (*CTLA4*, *FCGR2A*, *FCRL3*, *SH2B3*, *MAPKAPK5-AS1*, *ALDH2*, *TMEM116*, *ADAM1B*, *RP11-367J7.2*, *IL6R*, *NCR3*, *FCGR3B*, *VCAM1*) exhibiting significant associations with RA risk. They corresponded to four prioritized proteins: WASL (*CTLA4*), POLR2F (*FCGR2A*, *FCRL3*, *SH2B3*, *MAPKAPK5-AS1*, *ALDH2*, *TMEM116*, *ADAM1B*), IGSF11 (*FCRL3*, *RP11-367J7.2*), ADPGK (*IL6R*, *NCR3*, *FCGR3B*, *VCAM1*, *SH2B3*, *MAPKAPK5-AS1*, *ALDH2*, *TMEM116*, *ADAM1B*). Hand-curated searches revealed that the pairs of WASL*-CTLA4*, POLR2F*-FCGR2A*, POLR2F*-SH2B3*, and ADPGK*-ALDH2* shared concordant functional pathways, indicating vertical pleiotropy. No shared functional pathways were found in other pairs, suggesting the presence of potential horizontal pleiotropy (Appendix A).

### 3.7. Genetically Engineered Mouse Models of Prioritized Protein-Coding Genes

Genetically engineered mouse models for mouse orthologs of *NFKBIE*, *CCL21*, *ICOSLG*, *ALDH2*, *FCGR3A*, *IL6R*, *CD28*, *FCGR2A*, *TNFAIP3*, *CD40*, *FLT3*, *IFNGR2*, *MFAP2*, *SPRED2*, *SUGP1*, *CCL19*, *PADI4*, and *WASL* yielded immune cell abnormalities, encompassing cell number, morphology, and function, as well as immune organ defects, substantiating the intrinsic role of these genes in immune regulation. Additional phenotypic effects of these gene modifications are presented in Appendix A.

### 3.8. Cell-Type-Specific Expression of Prioritized Proteins in Synovium

Following the preprocessing of scRNA-seq data, cells were clustered and classified into four principal types (B cells, fibroblasts, monocytes, T cells) (Figure 6A). Cell composition varied between osteoarthritis (OA) and RA samples, with RA tissues showing a relative enrichment of B cells, fibroblasts, and T cells compared to OA (Figure 6B). All prioritized protein-coding genes, except HAPLN4, exhibited detectable expression within synovium, with distinct cell-type distributions and expression intensities (Figure 6C, Appendix A). Among these, the expressions of 14 genes (*ICOSLG*, *ALDH2*, *FCGR3A*, *IL6R*, *CD28*, *FCGR2A*, *PAM*, *TNFAIP3*, *CD40*, *IFNGR2*, *FCRL3*, *OLFML3*, *CELF2*, *H2AZ1*) were significantly enriched in specific cell types (Appendix A). Intergroup comparisons highlighted four genes (*FCGR2A*, *PAM*, *TNFAIP3*, *OLFML3*) with significant differential expression in their enriched cell types between RA and OA groups. The expressions of *FCGR2A* in monocytes and *OLFML3* in fibroblasts were elevated in RA versus OA, whereas *PAM* in fibroblasts and *TNFAIP3* in T cells were downregulated in RA relative to OA (Figure 6D, Appendix A).

### 3.9. Druggable Evidence for Nine Prioritized Proteins

Of the 32 prioritized proteins, 9 (ICOSLG, ERBB2, ALDH2, FCGR3A, IL6R, CD28, CD40, FLT3, and IFNGR2) were identified as targets of existing drugs that are either approved or undergoing clinical trials. Drugs aimed at ERBB2, FCGR3A, and FLT3 are primarily used in treating various cancers and benign tumors, while ALDH2 is targeted in therapies for a range of systemic diseases, notably cancer and substance dependence. Agents targeting ICOSLG, IL6R, and CD28 are broadly applied in managing immune system diseases. Additionally, CD40-targeting drugs have therapeutic roles in both cancer and immune diseases, depending on their specific pharmacological actions. Among these, medications targeting ERBB2 (Fostamatinib), IL6R (Levilimab, Sarilumab, Tocilizumab, Vobarilizumab), CD28 (TGN-1412), CD40 (Iscalimab, BI-655064), and FLT3 (Pexidartinib) have been approved or are in clinical trials for RA treatment. We further compared the Pro-MR-indicated effect directions of target proteins on RA and the pharmacological actions of existing drugs for these targets. The comparisons confirmed that the pharmacological actions of drugs targeting ICOSLG, ERBB2, ALDH2, CD28, CD40, FLT3, and IFNGR2 met the proposed effect directions from Pro-MR (Appendix A).

### 3.10. Phe-MR Identified Potential Side Effects of Nine Prioritized Proteins

Phe-MR analysis revealed additional associations between prioritized proteins and various disease traits. Certain prioritized proteins were associated with risks of other autoimmune diseases. For example, genetically predicted plasma levels of ALDH2, BCL2L15, WASL, POLR2F, and ADPGK were positively associated with hypothyroidism risk, while H2BC21 and H2BC26 showed negative associations. Genetic susceptibility to a higher FCGR2A level was linked to a reduced risk of ulcerative colitis, whereas a genetically predicted higher PAM level was associated with an increased risk of chronic ulcerative colitis. In another aspect, genetic susceptibility to higher levels of ALDH2, POLR2F, and ADPGK were associated with increased risks of various cardiovascular diseases. Furthermore, POLR2F and ADPGK also demonstrated potential roles as suppressors of certain malignancies (Appendix A).

### 3.11. Six Prioritized Proteins Partially Mediate the Effects of Modifiable Factors on RA

Among the 247 modifiable factors analyzed, 10 demonstrated significant associations with both RA and certain prioritized proteins, including 7 gut microbiome, 1 body measurement, 1 lifestyle, and 1 dietary pattern protein. Using multivariate MR (MVMR), we adjusted for these modifiable factors to determine the independent effects of prioritized proteins on RA. From this, we identified 26 modifiable factor-protein–RA pairs with potential mediation effects, for which mediation effect sizes were calculated (Appendix A). Among these, six pairs showed statistically significant mediation effects, as illustrated in Appendix A. For instance, the promoting effect of *Eubacterium brachy* on RA was mediated by the genetically predicted plasma SUGP1 level at 14.110%. The protective effect of *Holdemania* on RA was mediated by plasma levels of IGSF11 and H2AC25 at 51.724% and 35.632%, respectively. Additionally, the positive association between BMI and RA risk was mediated by plasma levels of POLR2F, ALDH2, and FCRL3 at 37.824%, 23.316%, and 2.073%, respectively.

## 4. Discussion

Leveraging the human plasma proteome, we identified and prioritized 32 proteins as potential therapeutic targets, followed by detailed validation and comprehensive characterization. Among all putative targets identified in this study, half were newly proposed, highlighting significant potential for future research and therapeutic development, and deepening insights for therapeutic discovery and mechanistic exploration in RA. In the following discussion, we concentrate on the high-reliability proteins validated through our analyses.

Multiple high-reliability prioritized proteins have been corroborated in previous genetic or basic studies to be associated with RA, including NFKBIE, ICOSLG, FCGR3A, CD40, FCRL3, and PADI4. Such consistency enhances the robustness of our findings.

NFKBIE binds to nuclear factor kappa-B (NFKB) to form an inactive cytoplasmic complex that prevents its nuclear translocation, maintaining the suppression of immune and inflammatory signaling pathways in the resting state [33]. Experimental studies showed that mice with NFKBIE gene defects exhibit a B cell expansion phenotype [33]. And non-synonymous SNPs in the *NFKBIE* gene region have been associated with RA susceptibility [34]. ICOSLG, FCGR3A, CD40, and FCRL3 are cell surface receptors or associated ligands. These proteins have consistently been identified as RA risk factors. Multiple GWASs have linked SNPs in *FCGR3A* [35], *CD40* [23], and *FCRL3* [36] with RA susceptibility. Elevated expression of *ICOSLG* [37], *CD40* [38], and *FCRL3* [39] has been observed in RA patients or inflammatory arthritis animal models. Downregulation of *ICOSLG* and *CD40* or blocking of their signaling suppressed immune responses and effectively mitigated joint damage in animal models [40,41], while *FCRL3* upregulation led to B cell tolerance disruption [42]. Notably, our reverse MR analysis revealed an inverse causal relationship between RA and plasma FCRL3, implying a potential circulatory progression mechanism in RA. So far, CD40-targeting drugs (Iscalimab, BI-655064) have been applied in RA clinical trials [43,44]. In addition, an ICOSLG inhibitor, AMG-557, has been in clinical trials for SLE and lupus arthritis, preliminarily showing safety and potential efficacy [45], with promising potential for RA. PADI4, predominantly expressed in neutrophil lineages, catalyzes the conversion of arginine to citrulline. Dysregulated PADI4 activation, leading to excessive citrullination, is believed to breach immune tolerance, resulting in the production of high-titer anti-citrullinated peptide antibodies in RA sera [46]. Genetic studies across multiple ethnicities have validated PADI4 as a susceptibility locus for RA [47]. Our PPI analysis reveals interactions between PADI4 and established RA drug targets such as TNF and CTLA4.

This study also proposed novel target proteins not previously mentioned. Of these, high-reliability cis-associated proteins include ALDH2, MFAP2, HAPLN4, and SUGP1, with Pro-MR suggesting that genetically predicted plasma levels of these proteins all correlate positively with RA risk. ALDH2 efficiently oxidizes toxic acetaldehyde into acetate, a key step in alcohol metabolism [48]. A study suggested ALDH2 as a citrullinated antigen in RA synovial tissue, potentially contributing to autoimmunity [49]. Anti-ALDH2 antibodies also served as biomarkers for Graves’ ophthalmopathy progression [50]. Interestingly, while ALDH2 is commonly associated with cardio-protection against oxidative stress [51], our Phe-MR analysis links a higher plasma ALDH2 level to increased cardiovascular risk. We presume that a genetically determined high ALDH2 level may correlate with greater lifetime alcohol exposure [52], thus driving the effect of genetically predicted ALDH2 on CVD risk. Disulfiram, an ALDH2 inhibitor, is currently approved, holding potential for immune-related diseases [53]. MFAP2, an extracellular matrix (ECM) glycoprotein, interacts with fibrillin to support microfibril function [54]. One of its essential roles is binding and sequestering active TGF-β within ECM [54]. *Mfap2* (−/−) mice exhibit heightened TGF-β activity [55]. This regulation of TGF-β signaling may bridge the effect of MFAP2 on RA. HAPLN4, also known as Bral2, is exclusively expressed in specific brain regions, such as the cerebellum and brainstem [56]. Its interaction with brevican is critical for synaptic stability and perineuronal net integrity [57]. *HAPLN4* has been discovered to be associated with psychiatric disorders [58]. The association between plasma HAPLN4 levels and RA risk may imply the intriguing interaction between the central nervous system and systemic autoimmune disorders. SUGP1 significantly modulates cholesterol metabolism via post-transcriptional regulation of *HMGCR* [59]. Overexpression of *SUGP1* has been reported to lead to an elevated plasma cholesterol level in vivo [59]. Our findings linking SUGP1 to RA reinforce the role of cholesterol metabolism in RA. In observational studies, however, RA patients’ lipid profiles exhibited complex, dynamic shifts influenced by inflammation, medication, and nutrition, sometimes even resulting in a “lipid paradox” [60]. Molecular research may aid in elucidating cholesterol’s causative role and associated pathways in RA.

However, it is noteworthy that some proteins exhibited effects in Pro-MR that contradict the general consensus, such as IL6R, FCGR2A, TNFAIP3, and IFNGR2. This discrepancy may be attributed to two factors. First, pQTLs used as IVs may introduce measurement artifacts when they also function as protein-altering variants [17]. For example, the pQTL (rs1801274) proxying the FCGR2A plasma level is a non-synonymous variant that not only influences the concentration of FCGR2A but also modifies its Fc binding affinity [61], thus complicating the interpretation of the protein effect. In another aspect, there were divergent proteoforms within the same protein which possessed heterogeneous spatial distributions and functions [23]. For example, membrane-bound IL6R and soluble IL6R, generated through alternative splicing or proteolytic processing, mediate class and trans signaling, respectively. The former pathway supports cytokine and anti-inflammatory functions, while the latter promotes pro-inflammatory effects in contrast [62]. It should be clarified that current proteomic techniques have limitations in distinguishing these proteoforms.

The inclusion of trans-pQTLs as IVs to elucidate genetic disease associations is a double-edged sword. While susceptible to horizontal pleiotropy, it can also reveal mediating genes distant from disease signals, thereby enriching our understanding of complex disease mechanisms. Among the exclusively trans-associated proteins, WASL, POLR2F, IGSF11, and ADPGK yielded high reliability. For instance, WASL, broadly explored in the oncology field, was associated with all RA-related phenotypes. The trans-pQTL (rs3087243) associated with WASL also acts as a cis-eQTL for *CTLA4*, whose expression is similarly causally linked to RA risk. Further analysis revealed vertical pleiotropy between WASL and *CTLA4*, with both involved in immune pathways. The trans-nature suggests that WASL is more likely downstream of *CTLA4* [23], shedding light on the potential biological pathway by which WASL influences RA. Likewise, prioritized protein-secondary gene pairs of POLR2F*-FCGR2A*, POLR2F*-SH2B3*, and ADPGK*-ALDH2* presented comparable mechanistic links. Moreover, POLR2F, IGSF11, and ADPGK were in horizontal pleiotropy with multiple genes or proteins at the corresponding pQTL. Although this may challenge MR assumptions, it has revealed additional cis-associated pathogenic candidates, offering expanded objectives for mechanistic investigation.

This study has limitations. First, the proteomic samples were predominantly derived from European-ancestry individuals, which limits the generalizability of the findings across ethnic groups. Second, horizontal pleiotropy may have influenced MR analysis despite the use of cis-pQTLs as IVs. Due to the limited IVs, alternative methods were precluded to test the robustness of the findings under potential violations of MR assumptions. Third, we did not identify significant associations between RA and certain proteins previously established, such as CD80, CTLA4, and JAK2. Some of them may lie beyond the coverage of current proteomic technologies, while some may be omitted under our stringent significance threshold, which is set to minimize type I errors from extensive multiple testing. Fourth, the absence of available eQTL data for certain proteins may lead us to underestimate their reliability. For instance, no eQTL data were accessible for CCL21 across all tissue datasets. Finally, as a largely computational study, these associations warrant further validation in clinical and basic research.

## 5. Conclusions

In conclusion, this study systematically prioritized 32 plasma proteins causally associated with rheumatoid arthritis using Pro-MR. These proteins underwent comprehensive validation, including MR meta-analysis, colocalization, and transcriptome-wide SMR, and were further stratified by reliability. Functional characterization revealed that several prioritized proteins are enriched in immune-related pathways, interact with established RA drug targets, exhibit abnormal immune phenotypes in genetically engineered mouse models, and show cell-type-specific expression in synovial tissues. In addition, a subset of these proteins demonstrated druggability and broader phenotypic associations. Collectively, these findings provide a strong foundation for the development of novel therapeutic strategies and offer new insights into the pathogenesis of RA. Further clinical and experimental studies are warranted to validate and translate these putative targets into therapeutic applications.

## Figures and Tables

**Figure 1 biomedicines-13-01841-f001:**
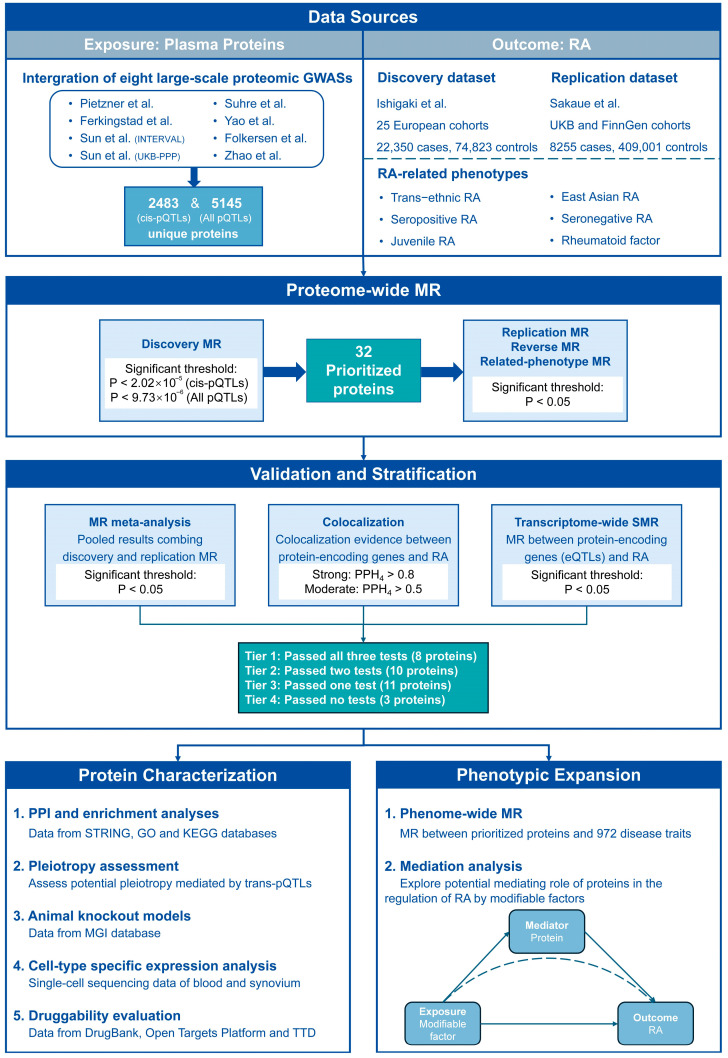
Study outline flowchart (Data sources: [16,17,18,19,20,21,22,23,24,25]). Abbreviations: GWAS, genome-wide association study; pQTL, protein quantitative trait locus; RA, rheumatoid arthritis; MR, Mendelian randomization; PPH_4_, posterior probability of H4; SMR, summary-data-based MR; eQTL, quantitative trait locus; PPI, protein–protein interaction; STRING, Search Tool for Recurring Instances of Neighboring Genes; GO, Gene Ontology; KEGG, Kyoto Encyclopedia of Genes and Genomes; MGI, Mouse Genomics Informatics; TTD, Therapeutic Target Database.

**Figure 2 biomedicines-13-01841-f002:**
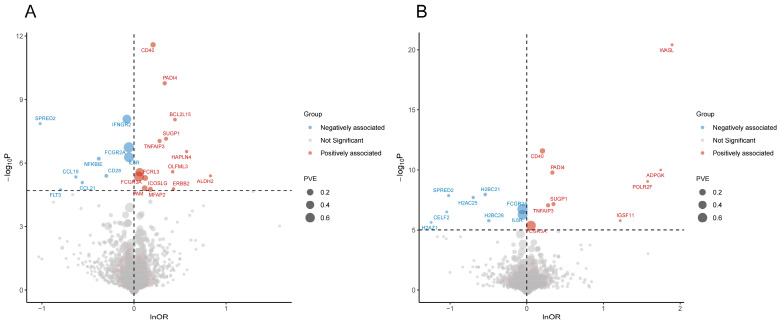
Volcano plots showing discovery-stage results of Pro-MR. (**A**) Results based on cis-pQTLs dataset. (**B**) Results based on all-pQTLs dataset. Abbreviations: OR, odds ratio; PVE, proportion of variance explained; Pro-MR, proteome-wide Mendelian randomization; pQTL, protein quantitative trait locus.

**Figure 3 biomedicines-13-01841-f003:**
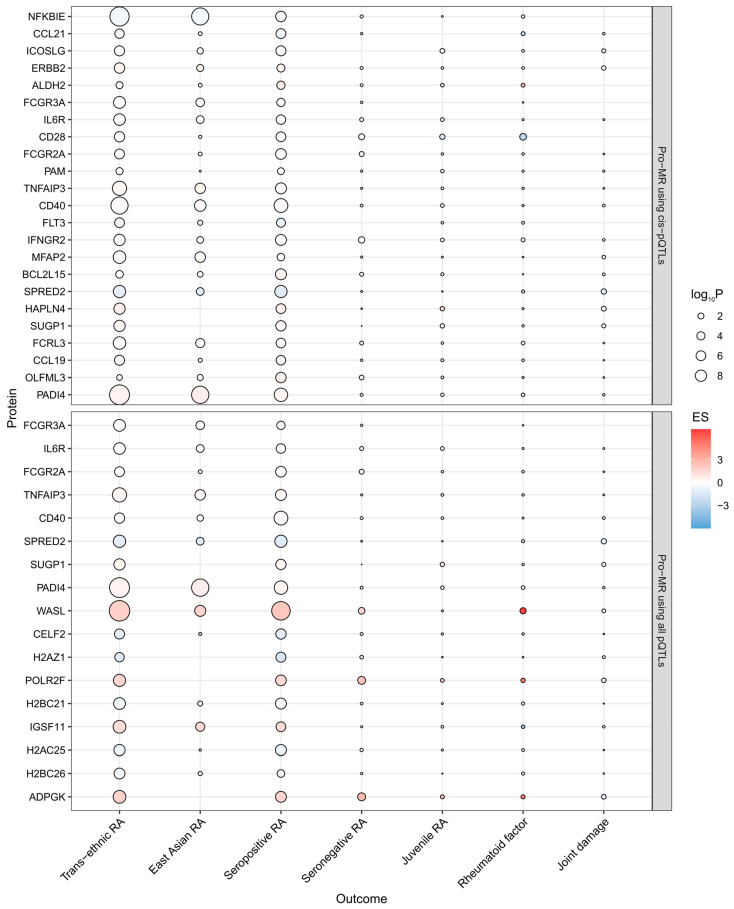
Bubble plot showing associations between plasma levels of prioritized proteins and RA-related phenotypes. Abbreviations: Pro-MR, proteome-wide Mendelian randomization; pQTL, protein quantitative trait locus; ES, effect size; RA, rheumatoid arthritis.

**Figure 4 biomedicines-13-01841-f004:**
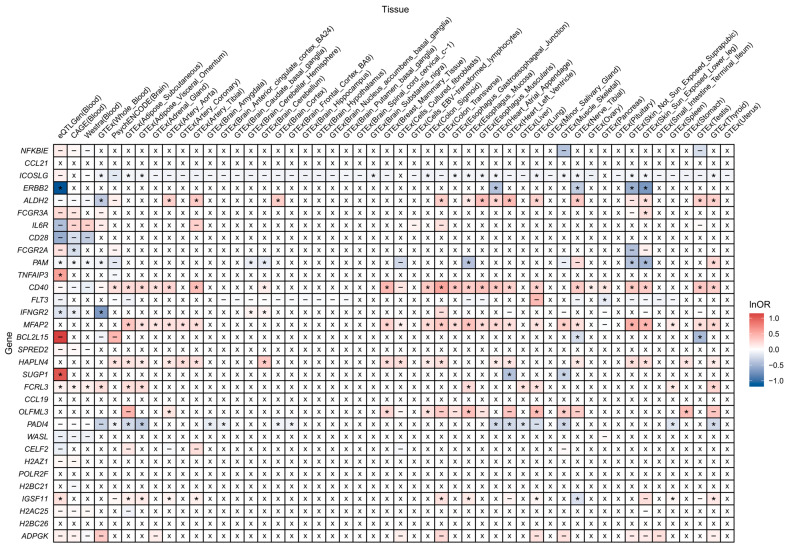
Heatmap displaying associations between expression levels of prioritized protein-coding genes and RA risk. Markers: *, meets the significance threshold (*p* < 0.050) and passes the HEIDI test; −, does not meet the significance threshold or fails the HEIDI test; ×, data unavailable. Abbreviations: CAGE, Consortium for the Architecture of Gene Expression; GTEx, Genotype-Tissue Expression; OR, odds ratio; RA, rheumatoid arthritis.

**Figure 5 biomedicines-13-01841-f005:**
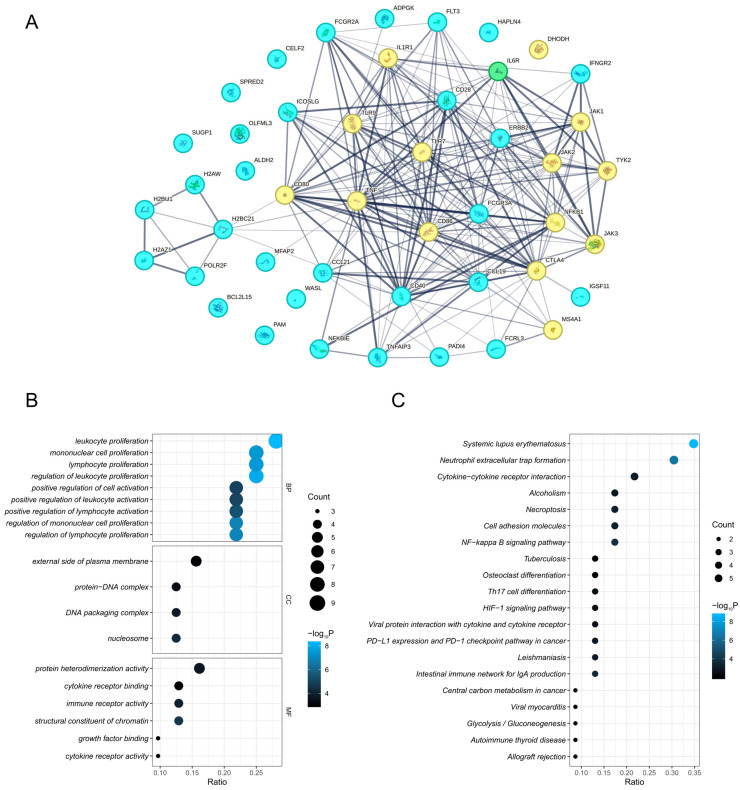
PPI and enrichment results. (**A**) PPI network. (**B**) GO enriched terms. (**C**) KEGG enriched pathways. In the PPI network, blue dots represent prioritized proteins in this study; yellow dots represent established drug targets for RA; green dots represent proteins that are both prioritized in this study and are established drug targets. H2AW and H2BU1 are alternative names for H2AC25 and H2BC26, respectively. Abbreviations: BP, biological process; CC, cellular component; MF, molecular function; PPI, protein–protein interaction; GO, Gene Ontology; KEGG, Kyoto Encyclopedia of Genes and Genomes.

**Figure 6 biomedicines-13-01841-f006:**
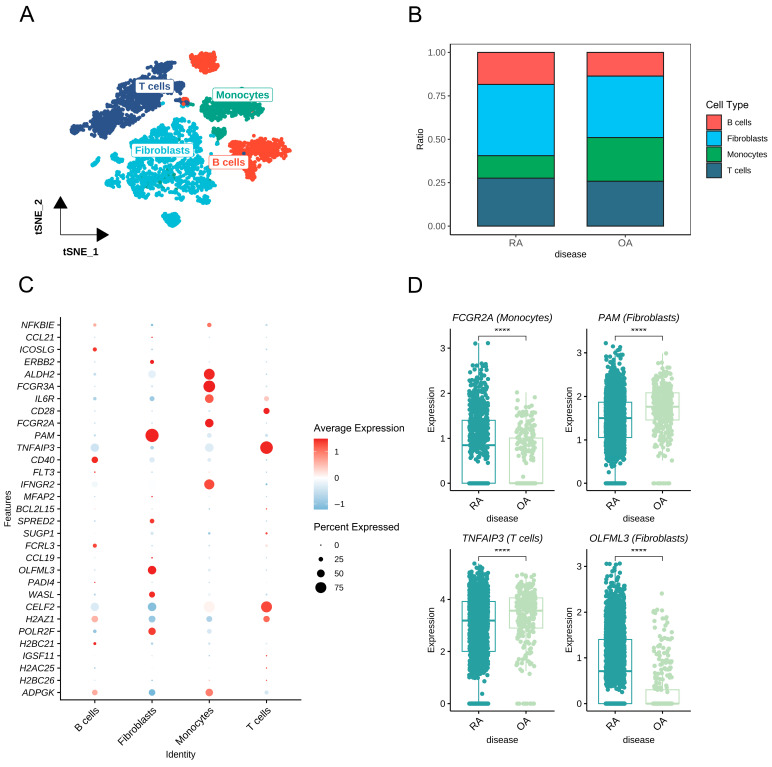
Cell-type-specific expression analysis of prioritized protein-coding genes in synovium using scRNA-seq data. (**A**) Identification and annotation of four distinct synovial cell clusters. (**B**) Composition of synovial cell types in RA and OA samples. (**C**) Bubble plot showing expression profiles of prioritized protein-coding genes in specific synovial cell types. (**D**) Intergroup comparisons reveal significant differences in expression levels of *FCGR2A*, *PAM*, *TNFAIP3*, and *OLFML3* in their respective enriched cell types (noted in parentheses) between RA and OA groups. Abbreviations: RA, rheumatoid arthritis; OA, osteoarthritis. ****, *p* < 0.0001.

**Table 1 biomedicines-13-01841-t001:** Information, validation, and stratification of prioritized proteins.

Protein Short Name ^a^	Uniprot ID	Full Name	MR Meta-Analysis		Colocalization	SMR	Tier
			OR	*p* Value			
NFKBIE	O00221	NF-kappa-B inhibitor epsilon	0.747	3.560 × 10^−17^	+	−	2
CCL21	O00585	C-C motif chemokine 21	0.654	9.605 × 10^−6^	−	−	3
ICOSLG	O75144	ICOS ligand	1.095	1.286 × 10^−5^	+	+	1
ERBB2	P04626	Receptor tyrosine-protein kinase erbB-2	1.320	5.989 × 10^−2^	−	+	3
ALDH2	P05091	Aldehyde dehydrogenase, mitochondrial	1.804	3.840 × 10^−6^	+	+	1
FCGR3A	P08637	Low-affinity immunoglobulin gamma Fc region receptor III-A	1.052	7.118 × 10^−10^	−	+	2
IL6R	P08887	Interleukin-6 receptor subunit alpha	0.958	2.113 × 10^−8^	+	−	2
CD28	P10747	T cell-specific surface glycoprotein CD28	0.795	5.446 × 10^−6^	−	−	3
FCGR2A	P12318	Low-affinity immunoglobulin gamma Fc region receptor II-a	0.955	2.933 × 10^−9^	+	+	1
PAM	P19021	Peptidyl-glycine alpha-amidating monooxygenase	1.054	4.011 × 10^−1^	−	+	3
TNFAIP3	P21580	Tumor necrosis factor alpha-induced protein 3	1.275	2.751 × 10^−10^	−	+	2
CD40	P25942	Tumor necrosis factor receptor superfamily member 5	1.174 (cis-pQTL) 1.191 (all pQTLs)	7.047 × 10^−4^ (cis-pQTL) 1.007 × 10^−12^ (all pQTLs)	+	+	1
FLT3	P36888	Receptor-type tyrosine-protein kinase FLT3	0.602	8.046 × 10^−2^	−	+	3
IFNGR2	P38484	Interferon gamma receptor 2	0.943	9.349 × 10^−4^	+	+	1
MFAP2	P55001	Microfibrillar-associated protein 2	1.172	2.050 × 10^−9^	−	+	2
BCL2L15	Q5TBC7	Bcl-2-like protein 15	1.290	1.829 × 10^−1^	−	+	3
SPRED2	Q7Z698	Sprouty-related, EVH1 domain-containing protein 2	0.474	7.533 × 10^−3^	−	−	3
HAPLN4	Q86UW8	Hyaluronan and proteoglycan link protein 4	1.478	4.386 × 10^−2^	+	+	1
SUGP1	Q8IWZ8	SURP and G-patch domain-containing protein 1	1.252	7.287 × 10^−2^	+	+	2
FCRL3	Q96P31	Fc receptor-like protein 3	1.055	1.256 × 10^−8^	+	+	1
CCL19	Q99731	C-C motif chemokine 19	0.618	3.105 × 10^−6^	−	−	3
OLFML3	Q9NRN5	Olfactomedin-like protein 3	1.230	3.293 × 10^−1^	−	+	3
PADI4	Q9UM07	Protein-arginine deiminase type-4	1.298	2.813 × 10^−4^	+	+	1
WASL	O00401	Actin nucleation-promoting factor WASL	4.424	2.875 × 10^−4^	+	−	2
CELF2	O95319	CUGBP Elav-like family member 2	0.565	2.405 × 10^−1^	−	−	4
H2AZ1	P0C0S5	Histone H2A.Z	0.491	1.975 × 10^−1^	−	−	4
POLR2F	P61218	DNA-directed RNA polymerases I, II, and III subunit RPABC2	4.250	1.693 × 10^−13^	+	−	2
H2BC21	Q16778	Histone H2B type 2-E	0.635	3.379 × 10^−8^	−	−	3
IGSF11	Q5DX21	Immunoglobulin superfamily member 11	2.812	1.081 × 10^−7^	−	+	2
H2AC25	Q7L7L0	Histone H2A type 3	0.646	1.039 × 10^−1^	−	−	4
H2BC26	Q8N257	Histone H2B type 3-B	0.654	3.602 × 10^−6^	−	−	3
ADPGK	Q9BRR6	ADP-dependent glucokinase	4.840	3.853 × 10^−14^	+	−	2

^a^, Protein short name was reported using the non-italicized form of the corresponding gene symbol; +, supported by the test; −, not supported by the test. The parentheses indicate the dataset used. Abbreviations: OR, odds ratio; MR, Mendelian randomization; SMR, summary-data-based Mendelian randomization.

## Data Availability

The data utilized in this study are either publicly available or accessible upon reasonable request, with specific sources detailed in the manuscript and Appendix A.

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
