# Peer review of "Investigation of Novel Therapeutic Targets for Rheumatoid Arthritis Through Human Plasma Proteome"

_biomedicines, 2025, doi:10.3390/biomedicines13081841_

Round 1
Reviewer 1 Report
Comments and Suggestions for Authors
- Line 47-49: Please provide relevant references. It seems Reference 5 says MTX monotherapy.
- Please check whether ‘Ishikagi’ in Figure 1 would be ‘Ishigaki’ which could be found as the first author of reference 24.
Author Response
Comments 1: Line 47-49: Please provide relevant references. It seems Reference 5 says MTX monotherapy.
Response 1: Thank you for pointing this out. This part of the text is also based on reference 5, which states: "Each drug has limited efficacy and achieves low disease activity in up to 40% and remission in up to 20% of patients with RA (Table 2)." To avoid ambiguity, we have now added the citation at the end of the sentence. The revised text reads: "However, there are still several limitations to the existing disease-modifying antirheumatic drugs (DMARDs). First, each drug has limited efficacy, with monotherapy achieving disease activity reduction or remission in only up to 40% of RA patients [5]."
Comments 2: Please check whether ‘Ishikagi’ in Figure 1 would be ‘Ishigaki’ which could be found as the first author of reference 24.
Response 2: Thank you for your careful correction. The correct name should be "Ishigaki." This was our oversight, and it has been corrected in the revised Figure 1.
Reviewer 2 Report
Comments and Suggestions for Authors
- The abstract mentions integrating data from eight proteomic GWASs but does not specify whether these were overlapping or independent cohorts. A brief note on harmonization methods would strengthen reproducibility.
- Proteome-wide Mendelian randomization (Pro-MR)" is introduced without elaboration. Consider briefly defining Pro-MR (e.g., "a causality-testing framework leveraging genetic variants as instrumental variables") for broader accessibility.
- The "4 reliability tiers" are mentioned but not defined. A short phrase (e.g., "based on colocalization evidence and MR robustness") would help readers assess stratification rigor.
- While 18 genes showed immune phenotypes in mice, were these phenotypes directly relevant to RA (e.g., synovitis, autoantibody production)? Highlighting this would bolster translational relevance.
- The nine prioritized proteins linked to existing drugs could be contextualized (e.g., "including TNF inhibitors or JAK/STAT modulators") to emphasize clinical applicability.
- The abstract notes effects on "other phenotypes" but omits examples. Specify key findings (e.g., "comorbid cardiovascular traits") to showcase broader implications.
- The conclusion could explicitly state how many targets are novel (vs. overlapping with known RA pathways) to highlight originality.
- A minor caveat (e.g., "potential bias from plasma protein levels not reflecting synovial tissue expression") would demonstrate methodological awareness.
Author Response
Comments 1: The abstract mentions integrating data from eight proteomic GWASs but does not specify whether these were overlapping or independent cohorts. A brief note on harmonization methods would strengthen reproducibility.
Response 1: Thank you for your valuable comment. The cohorts underlying the eight large-scale proteomic GWASs are independent and do not overlap. We supplemented the clarification of this point in page 9, lines 99–100. Due to space limitations, further details on data integration have been provided in Supplementary File 2, Data Integration of Plasma Proteins.
Comments 2: Proteome-wide Mendelian randomization (Pro-MR)" is introduced without elaboration. Consider briefly defining Pro-MR (e.g., "a causality-testing framework leveraging genetic variants as instrumental variables") for broader accessibility.
Response 2: Thank you for pointing this out. We have added the relevant information in page 2, lines 70–74.
Comments 3: The "4 reliability tiers" are mentioned but not defined. A short phrase (e.g., "based on colocalization evidence and MR robustness") would help readers assess stratification rigor.
Response 3: Thank you for your comment. The tiers have already been defined in page 5, lines 153–156.
Comments 4: While 18 genes showed immune phenotypes in mice, were these phenotypes directly relevant to RA (e.g., synovitis, autoantibody production)? Highlighting this would bolster translational relevance.
Response 4: Thank you for the reminder. Due to space limitations, detailed information on the number of genes involved in each phenotype is provided in Supplementary File 1, Table S10.
Comments 5: The nine prioritized proteins linked to existing drugs could be contextualized (e.g., "including TNF inhibitors or JAK/STAT modulators") to emphasize clinical applicability.
Response 5: Thank you for your suggestion. The druggable evidence for the prioritized proteins is described in detail on pages 14–15, lines 367–382, and is also listed in Supplementary File 1: Table S12. Target proteins with druggable evidence are further discussed in detail in the Discussion section.
Comments 6: The abstract notes effects on "other phenotypes" but omits examples. Specify key findings (e.g., "comorbid cardiovascular traits") to showcase broader implications.
Response 6: Thank you for your insight. The relevant content has been added on page 1, lines 21–23.
Comments 7: The conclusion could explicitly state how many targets are novel (vs. overlapping with known RA pathways) to highlight originality.
Response 7: Thank you for pointing this out. We have focused our discussion on the novel RA targets with higher levels of evidence on page 17, lines 452–480.
Comments 8: A minor caveat (e.g., "potential bias from plasma protein levels not reflecting synovial tissue expression") would demonstrate methodological awareness.
Response 8: Thank you for your insight. In consideration of this point, we have performed a cell-type-specific expression analysis in synovial tissue, as described on page 5, lines 182–187.
Reviewer 3 Report
Comments and Suggestions for Authors
This study aims to identify potential therapeutic targets for rheumatoid arthritis based on the human plasma proteome. It is a very interesting and useful paper.
The abstract and introduction are well organised and give all the needed information.
The methods part is in detail explained.
Results: Too many figures. My question is, is it necessary to introduce all 7 figures here? Could any of them be transferred to supplementary and keep only the most important ones?
Discussion: The conclusion part should be separately introduced with a subheading. In addition, the conclusion should introduce more concrete findings from this study.
Author Response
Comments 1: This study aims to identify potential therapeutic targets for rheumatoid arthritis based on the human plasma proteome. It is a very interesting and useful paper.
The abstract and introduction are well organised and give all the needed information.
The methods part is in detail explained.
Results: Too many figures. My question is, is it necessary to introduce all 7 figures here? Could any of them be transferred to supplementary and keep only the most important ones?
Discussion: The conclusion part should be separately introduced with a subheading. In addition, the conclusion should introduce more concrete findings from this study.
Response 1: Thank you for your recognition and valuable suggestions. The original Figure 7 has been moved to Supplementary File 3 and renamed as Figure S3. In addition, we have added a subheading to the Conclusion section and expanded its content accordingly [page 17-18, lines 517-529].
Reviewer 4 Report
Comments and Suggestions for Authors
Dear Authors!
Thank you for the opportunity to review your manuscript
Rheumatoid arthritis is the most frequent inflammatory arthropathy in adults with a high risk of joint disability, invalidisation, and decreased life quality.
Despite the rising number of new drugs, the treatment of RA patients remains challenging, and new data about the disease pathogenesis, biomarkers, and treatment targets are needed.
The authors provided the study, and protein quantitative trait loci were extracted and integrated from eight large-scale proteomic GWASs. Proteome-wide Mendelian randomization (Pro-MR) was performed to prioritize proteins causally associated with RA.
In the introduction section, the Authors provide the study's relevance, but the first part of the Introduction contains common information about RA and could be shortened to focus on the study's goals. The manuscript is big, and some shortening is required.
In the Methods the Authors provided the deteiled axpalnation of the study's components and provided the flow-chart of the study
The results are confirmed with graphs and tables
The discussion contains a review of the relevant literature.
The discussion has a Limitations section, and the Authors disclosed the weak parts of the study
The conclusion summarizes the study's results, but it is brief and somewhat unclear. Please focus on the main study findings and futitre perspetives
Author Response
Comments 1: Dear Authors!
Thank you for the opportunity to review your manuscript
Rheumatoid arthritis is the most frequent inflammatory arthropathy in adults with a high risk of joint disability, invalidisation, and decreased life quality.
Despite the rising number of new drugs, the treatment of RA patients remains challenging, and new data about the disease pathogenesis, biomarkers, and treatment targets are needed.
The authors provided the study, and protein quantitative trait loci were extracted and integrated from eight large-scale proteomic GWASs. Proteome-wide Mendelian randomization (Pro-MR) was performed to prioritize proteins causally associated with RA.
In the introduction section, the Authors provide the study's relevance, but the first part of the Introduction contains common information about RA and could be shortened to focus on the study's goals. The manuscript is big, and some shortening is required.
In the Methods the Authors provided the deteiled axpalnation of the study's components and provided the flow-chart of the study
The results are confirmed with graphs and tables
The discussion contains a review of the relevant literature.
The discussion has a Limitations section, and the Authors disclosed the weak parts of the study
The conclusion summarizes the study's results, but it is brief and somewhat unclear. Please focus on the main study findings and futitre perspetives
Response 1: Thank you for your comment. Based on your suggestion, we have reorganized the Conclusion section accordingly [pages 17–18, lines 517–529].